# Microbial communities and isotopes as novel tracers for groundwater flow paths in the multi-layered aquifer system in Kurikka, western Finland

5 Lotta Purkamo<sup>1</sup>, Juuso Ikonen<sup>1</sup>, Marie-Amélie Pétré<sup>1</sup>, Niko Putkinen<sup>1</sup>, Minna Myllyperkiö<sup>1</sup>, Anna-Maria Hokajärvi<sup>2</sup>, Tarja Pitkänen<sup>2,3</sup>, Ilkka T. Miettinen<sup>2</sup>

<sup>1</sup>Geological Survey of Finland (GTK), Espoo, 02150, Finland

<sup>2</sup>Finnish Institute for Health and Welfare (THL), Department of Public Health, Kuopio, 70210, Finland

<sup>3</sup>University of Helsinki, Faculty of Veterinary Medicine, Helsinki, 00014, Finland

Correspondence to: Lotta Purkamo (lotta.purkamo@gtk.fi)

**Abstract.** Groundwater is a critical resource supplying nearly half of the world's drinking water. This study focuses on the Kurikka buried valley aquifer system in western Finland, characterized by complex hydrogeology dictated by the bedrock topography and sediment cover producing artesian conditions in deep aquifers. Using a multitracer approach, the study incorporates hydrogeochemical, isotopic (δ³4S, 87Sr/86Sr) and microbial community analyses with residence time indicators (CFCs, SF<sub>6</sub>, ³H, ³H/³He). Groundwater samples collected from 10 sites revealed differences in residence times, microbial diversity and community compositions, as well as large variation in the strontium and sulfur isotopic compositions. The bedrock groundwater sample revealed a more evolved water type, consistent with longer residence time, strong water-mineral interactions and typical deep subsurface bacterial members. Groundwater from the superficial unconsolidated aquifers contained a modern water component (

50

used to assess groundwater residence times, this study aims to go further by incorporating microbial community analyses along with strontium and sulfur isotopes. Previous studies have shown the potential usability of microbial community analysis in hydrogeological studies (Ben Maamar et al. 2015). Here we show that microbial data can provide novel insights into aquifer functioning within a buried valley aquifer system in western Finland, hosting significant water resources. The aim for the hydrogeological studies in the area is to find enough groundwater to support large-scale abstraction to meet the water demands of 150,000 residents and the industries in cities of Vaasa and Kurikka. The aquifer system has a complex groundwater recharge route originating from the high-standing areas west of the Kurikka buried valleys. From the recharge area, groundwater flows toward the buried valley aquifers, where a coarse-grained aquifer consists of the interplay between permeable sediments and fractured crystalline bedrock (Putkinen et al. in review, 2025). However, the flow paths and potential mixing of shallow and deeper groundwater, including possible input from the deep bedrock groundwater, are still poorly understood.

The hydrogen ( $\delta^2$ H) and oxygen ( $\delta^{18}$ O) isotopic composition of the water molecule can be used together with the isotopic compositions of selected solutes in the water sample to characterize water types, trace the origin of the water and to assess mixing of different waters and water pathways, as well as to extract more information on the chemical and biological processes in the study area (Carreira and Marques 2024, Ikonen et al. 2025, Luoma et al. 2024, Porru et al. 2024, Åberg et al. 2025). Changes in the isotopic compositions during the water cycle, from precipitation to groundwater recharge and eventual discharge, produce the tracer characteristics and form the basis for using isotopes as tracers in hydrogeological studies. With oxygen and hydrogen, the fractionation is caused by physical processes, i.e. evaporation, or mixing of different waters. The aim here was to use the water stable isotopes to determine the origin of water and residence times of the groundwater samples from the deep unconsolidated aquifer and the bedrock observation well and to assess the recharge periods for the groundwater.

Strontium (Sr) has four stable isotopes <sup>88</sup>Sr (82.53 %), <sup>87</sup>Sr (7.04 %), <sup>86</sup>Sr (9.87 %), and <sup>84</sup> Sr (0.56 %). As a result of radioactive decay of rubidium, the amount of the radiogenic isotope <sup>87</sup>Sr increases over geologic time. The stable and constant <sup>86</sup>Sr is used to determine the <sup>87</sup>Sr/<sup>86</sup>Sr ratio that is commonly used in hydrogeological studies (Shand et al. 2009). The isotopic composition of strontium analysed from a water sample corresponds to the local geochemistry. In Finland, the mineralogy of rock types usually produces an <sup>87</sup>Sr/<sup>86</sup>Sr value between 0.7 – 0.8 (Kaislaniemi 2011). The silicate rich rocks have higher potassium content (potassium feldspar, mica minerals) producing the higher <sup>87</sup>Sr/<sup>86</sup>Sr values, while carbonate rocks rich in calcium are associated with lower <sup>87</sup>Sr/<sup>86</sup>Sr values closer to the <sup>87</sup>Sr/<sup>86</sup>Sr value of seawater (0.709). The isotopic composition of strontium can be characterized as a conservative tracer, since there is no noticeable fractionation in chemical or biological reactions in low temperature/pressure environments (Bullen et al. 1996, Shand et al. 2009). Isotopic analysis of strontium was selected here to reveal information on the local mineralogy, and to see if information on groundwater mixing could be revealed and to set a precursor for further isotopic studies in the area.

The isotopic composition of sulphur (S), represented as  $\delta^{34}$ S, is the deviation of the ratio of its two isotopes  $^{32}$ S and  $^{34}$ S from the international standard. The range of the isotopic composition of sulphur is normally explained by the sulphur source

65

material (i.e. sulphide minerals), variation in redox conditions and microbial activity. The most important fractionation process in low temperature unconsolidated environments is the microbial reduction of sulphur, where microbes remove the lighter  $^{32}$ S increasing the ratio in the dissolved sulphur of the water sample (Onac et al. 2011). Sulphide oxidation (reoxidation), in turn, produces lighter isotopic compositions of sulphur in water samples. The isotopic composition of sulphur from a water sample can be linked to the microbiological data (Harrison and Thode 1958, Kaplan and Rittenberg 1963. Samborska-Goik and Bottrell, 2025), due to microbes behaving in an isotopically selective manner, preferring the weaker light-isotope bonds. The lower energy needed to break the  $^{32}$ S-O bond produces the enrichment seen in the analysed water. The analysis of  $\delta^{34}$ S was used in this study to link the microbial communities to the isotopic compositions of sulphur in the groundwater samples.

As part of this multitracer approach, several groundwater residence time indicators were used to characterize the buried valley aquifer system: Tritium (<sup>3</sup>H), Tritium/Helium-3 (<sup>3</sup>H/<sup>3</sup>He), chlorofluorocarbons (CFCs), sulphur hexafluoride (SF<sub>6</sub>) and noble gases (Helium isotopes and Neon). These tracers are useful because of their known input histories or predictable decay or accumulation in the subsurface. While <sup>4</sup>He is useful for dating ancient groundwater, <sup>3</sup>H, <sup>3</sup>H/<sup>3</sup>He, CFCs and SF<sub>6</sub> are used to identify "modern" groundwater (typically less than 60 years old). They are used to trace groundwater flow paths and inform on the renewal rate and vulnerability of the resource, which are important in a perspective of sustainably developing the resource. These groundwater dating tools have been widely applied across a range of hydrogeological contexts, including in hard-rock aquifers (Cook and Solomon 1997; Jaunat et al. 2012; Visser et al. 2013; Åkesson et al. 2015; Meyzonnat et al. 2023 Osenbrück et al. 2006; Massmann et al. 2008;)

Groundwater as a dark, cool, and typically oligotrophic environment selects specific microbial communities compared to the aquatic ecosystems aboveground (Lee et al., 2018). Groundwater habitats vary in geological, chemical and hydrological properties. They can cover a small water body or an entire regional aquifer spanning hundreds of kilometres. The geochemistry of the aquifer may give clues on ongoing microbial metabolic processes, but because of the enormous versatility of still unknown microbes inhabiting groundwater systems, it should not be the only source of information (Flynn et al., 2013). Understanding the microbial contribution to ecosystem processes, such as cycling of nutrients and regulating the redox environment, will aid in interpreting the biogeochemical properties of aquifers. Microbes in nutrient-poor environments are considered to represent heterotrophs that are adapted to prevailing conditions. Heterotrophs break down organic matter and release CO<sub>2</sub> into groundwater. On the other hand, chemolithoautotrophic organisms can be another component of the microbial community, and their significance especially in deeper groundwater habitats is prevalent (Hutchins et al. 2016). These autotrophs can use CO<sub>2</sub> to produce methane in anaerobic conditions. Oxygen availability and pH play a significant role on sulphur and iron cycling as well. Sulphate and iron reduction typically require anoxic conditions, but if oxygen is available, thermodynamics dictate oxidation reactions in groundwater energetically more favourable. In acidic conditions, iron reducers have a significant advantage over sulphate reduction (Bethke et al., 2011). As most microbes are specialized in either reduction

or oxidation rather than being capable of both, the energy substrate availability will determine the microbial community structure to some extent.

Although modern molecular biological methods, such as massive parallel sequencing techniques have aided the environmental microbiologists in gathering information about the taxonomic and functional diversity of microbes, microbial community structure remains to be fully exploited to understand hydraulic connections in aquifers (Merino et al., 2022). The aim of this study was to test the potential of using a microbial fingerprint to distinguish between the origins of groundwater in different locations of the aquifer. More specifically, the study aims to characterise natural groundwater microbial community structures in various aquifer locations, conduct chemical and isotopic analyses of sampled water to evaluate the usability of these natural tracers, and apply groundwater residence time indicators to bedrock and buried valley aquifer samples. This multitracer approach in hydrogeochemistry combined with lesser used microbial community analysis provides a robust tool for assessing and identifying the connectivity and flow paths within complex multi-aquifer systems. This information aids in making informed decisions about groundwater management and potential contamination remediation.

# 1.1 Site description

The study site is situated in the Southern Osthrobothnia region in western Finland. Region is characterized by flat lying terrain subject to agriculture and food production existing 70 kilometres from the Bothnian Sea shore (Fig. 1). The groundwater system of the Kurikka buried valley is underlain by 1.88 Ga year-old Palaeoproterozoic metamorphosed granites, schists, and biotite paragneisses (Lahtinen et al. 2017, Ruuska et al. 2023). The early development of the Palaeoproterozoic bedrock during the Svecofennian orogeny with crustal movements, shaped the Kurikka area and led to the formation of protovalleys. Since then, the area has experienced a multi-stage burial-erosion development over a long period of time until it reached its current buried form (Hall et al. 2021). Currently, this ~1.5 Ga old and 70–120 m deep buried valley system is filled with highly variable Late Pleistocene sediments. During its most recent stages of development, the area that lay beneath the central part of the Fennoscandian Ice Sheet (FIS) during the Late Weichselian Glaciation (23,000-10,500 years ago, e.g, Stroeven et al., 2016), which covered southern Finland and contributed to the preservation of the older sediments due to minimal erosion (Putkinen et al. *in review* 2025). Specifically, the deepest sediments typically consist of deposits that have groundwater potential for municipal drinking water abstraction.

The regional hydrogeology of the 240,5 km<sup>2</sup> study area is strongly controlled by bedrock topography. Topographical differences, together with silt-clay sediment and basal till on the topmost part of the sediment cover, produce an artesian character to the deep aquifers (Fig 1). The general direction of groundwater flow is from the highland region towards the buried valley aquifers, and it is assumed that a geochemical evolution occurs along the flow path.

Figure 1: Left, a geological map of the study area and groundwater sampling locations; right: general cross-section of the Paloluoma buried valley (modified from Rashid 2022). AF- and AT- refers to aquifer and aquitard, respectively. AT 3-4-5-6: till, AF3-4: fine sand, AF2: sandy gravel, AF5: coarse sand. Vertical exaggeration: 15X.

# 2 Materials and Methods

In this hydrogeochemical study of the region, the sampling points were selected in each buried valley connected to the main Kyrönjoki Valley (Fig. 1). Additionally, the study included a bedrock borehole (R56) located beneath the Paloluoma valley and a point further north (NOPPA15) in an adjacent valley reflecting a more superficial esker system.

Table 1. Details of the sampling sites in Kurikka multilayer aquifer system.

| Sample ID | Sampling  | Location        | Total depth of | Screen depth interval, | Aquifer type           |
|-----------|-----------|-----------------|----------------|------------------------|------------------------|
|           | date      |                 | the well,      | m                      |                        |
|           |           |                 | m              |                        |                        |
| NOPPA 15  | 4.10.2021 | Nenättömänluoma | 15.6           | *6.6-9.1; 10.6-13.6    | Shallow unconsolidated |

| MIHP6   | 5.10.2021 | Paloluoma                | 77.7  | 50.9-76.9             | Deep unconsolidated |
|---------|-----------|--------------------------|-------|-----------------------|---------------------|
| MIHP15  | 5.10.2021 | Paloluoma                | 64.6  | 38.1-64.6             | Deep unconsolidated |
| R56     | 5.10.2021 | Paloluoma                | **162 | **                    | Bedrock             |
| LOHI30  | 5.10.2021 | Lohiluoma                | 78.4  | 50.4-64.4             | Deep unconsolidated |
| KUU19   | 6.10.2021 | Paloluoma                | 72.9  | 49.5-72.9             | Deep unconsolidated |
| НÄЈҰ30  | 6.10.2021 | Kyrönjoki main<br>valley | 17.3  | ***                   | Deep unconsolidated |
| MIETO17 | 6.10.2021 | Kyrönjoki main<br>valley | 63.8  | *29.8-48.8; 49.8-60.8 | Deep unconsolidated |
| HARJA10 | 7.10.2021 | Kyrönjoki main<br>valley | 60.8  | *37.8-39.8; 41.8-51.8 | Deep unconsolidated |
| НÄЈҮ11  | 7.10.2021 | Häjynluoma               | 59    | 39-59                 | Deep unconsolidated |

<sup>\*</sup> Two screens

The water samples were collected between 4th and 7th of October in 2021. The groundwater wells were purged appropriately, and the water levels were measured at each well before sampling. The field measurements for temperature (T), dissolved oxygen (DO), pH, electrical conductivity (EC) and oxygen reduction potential (ORP) were done with YSI multi-parameter sonde (YSI EXO1) (YSI Inc. Yellow Springs, OH, US). Alkalinity was measured in the field with a HACH digital titrator (HACH Company, Loveland, CO, US) and a cartridge of sulfuric acid (1.600N). For hierarchical clustering, selected parameters (pH, EC, T, DO and alkalinity) were used for calculating Euclidean distance matrix, and clustering was done with UPGMA (Unweighted Pair Group Method with Arithmetic Mean) method in R with vegan, ggplot2 and ggdendro packages (Oksanen et al., 2012, de Vries & Ripley, 2024, Wickham, 2016). Sample pretreatment for hydrogeochemistry and isotopes, filtering and acidification, was performed in the field after sampling. Water for isotopic analyses was filtered with a 0.45 μm Ca-S filter into acid-washed 250 ml bottles and acidified with nitric acid (5 ml per sample bottle). Treated samples were kept cool during field work and transportation to the laboratory. The chemical parameters from the water samples were analysed by the Eurofins Labtium laboratory. Dissolved element concentrations were analysed with ICP-MS and anion concentrations (Br-, Cl-, F, NO³- ja SO₄²-) with ion chromatography. The dissolved and total organic carbon (DOC, TOC) were also analysed at the Eurofins Labtium laboratory.

<sup>\*\*</sup> Bedrock borehole, no screen in the sediment layers. Total length of borehole in the table. Drilled with 60 °dip.

<sup>\*\*\*</sup> No screen. Continuously overflowing.




The isotopic compositions of oxygen, hydrogen, strontium and sulfur from the water samples were analysed in the GTK research laboratory in Espoo, Finland. The  $\delta^{18}O$  and  $\delta^{2}H$  were analysed with a Picarro L2130i CRDS (Cavity Ring Down Spectroscopy) analyser. With L2130i the analytical uncertainty for  $\delta^{18}O$  is < 0.1 % and for  $\delta^{2}H$  < 0.3 %.

For the <sup>87</sup>Sr/<sup>86</sup>Sr analysis, a concentration-dependent amount of prefiltered and -acidified samples were evaporated to dryness and dissolved with uc. 3.2 N HNO<sub>3</sub> for ion exchange. Strontium was eluted with uc. 0.05N HNO<sub>3</sub> acid by 100 μl of Sr-specific resin (TrisKem Sr Resin, 50–100 μm). For the measurement, the samples were diluted to a Sr concentration of approx. 20 ppb (uc. 2% HNO<sub>3</sub>). The analyses were carried out by using an Aridus 3 Desolvating system (DSN) with 50μl PFA ConcentricFlow nebulizer and a Multi-Collector Inductively Coupled Plasma Mass Spectrometer (MC-ICP-MS, Nu Instruments<sup>TM</sup>) at low mass resolution (Δm/m = 400). The isotopic measurements were performed in static mode using five faraday detectors, and 10 blocks of 6 integrations of approximately 8 s. The standard reference material NBS987, was used to monitor the precision and accuracy of the measurements at the beginning, and the end of every session and after every fifth sample. The obtained average of <sup>87</sup>Sr/<sup>86</sup>Sr was 0.710270 (± 0.000066, 2 sd, n=5), which is close to the reference value 0.710250 (± 0.00004, 2 sd, n=2306, GEOREM database, http://georem.mpch-mainz.gwdg.de/).

Prior to the δ³⁴S analysis, samples were eluted following the classic liquid column chromatography technique (Paris et al, 2013). Based on the S-concentration, part of the filtered and acidified sample was evaporated to dryness and dissolved with uc. 0.25 % HCl. Cations were removed with BioRad AG50X8 resin and samples were further diluted with uc. 2 % HNO₃ to approx. 0.25 ppm concentration. Sodium was added to the samples (0.5 ppm) before the analysis to improve the sensitivity of sulphur (Yu et al., 2017). The analyses were carried out using an Aridus 3 Desolvating system (DSN) with 50µl PFA
ConcentricFlow nebulizer and a MC-ICP-MS (Nu Instruments TM) at medium mass resolution (Δm/m = 3000). The S isotopic measurements were performed in static mode using three faraday detectors, and 3 blocks of 20 integrations of 8 s. The average value of the IAEA standard S-3 was - 32.0 (± 0.3, 1sd, n=5), while the recommended values is -32.2 (±0.4 ‰ 1sd, n=36, Georem database, http://georem.mpch-mainz.gwdg.de/).

Samples of groundwater for CFCs and SF<sub>6</sub> analysis were collected in two tins, each holding a ground glass flask. The opened can and flask were placed in a bucket. Then, the pipe from the pump was placed to the bottom of the flask, which was rinsed multiple times until the bucket overflowed. After the flask was closed with a plug and secured with a clamp, the can container was closed with a circlip (underwater). CFCs and SF<sub>6</sub> were analysed by gas chromatography with an electron capture detector (GC-ECD) at Dr. Oster's trace substances laboratory (Wachenheim, Germany). Samples for tritium analysis were collected in 1L HDPE bottle and stored in a cold room (5°C) until analysis. Tritium was analysed at Hydroisotop GmbH (Germany) using liquid scintillation spectroscopy LSC after electrolytical enrichment (Perkin Elmer Quantulus GCT 6220). Tritium concentrations are reported in tritium units (TU) with double standard deviation (1TU=0.119 Bq/L). Samples for noble gases analyses were collected in duplicate in clamped-off copper tubes connected to the pumping line. Noble gases (Helium isotopes and Neon) were analysed at the Bremen Mass Spectrometric Facility according to Sültenfuß et al. (2009).

The partial pressure of CO<sub>2</sub> (pCO<sub>2</sub>) was calculated with the software Diagrammes (Simler 2012) based on temperature, pH and HCO<sub>3</sub>.





Groundwater samples were also retrieved for microbial community structure analysis. As groundwater is a naturally oligotrophic environment, the microbial cell numbers were estimated to be low. Therefore, to collect enough microbial biomass for DNA metabarcoding analysis, 65 - 183 l of groundwater from each groundwater well was filtered by a dead-end ultrafiltration method (DEUF, ASAHI Rexeed-25A, Asahi Kasei Medical Co., Ltd., Tokyo, Japan) with a flow rate of 2-3 l/min using a sterile silicone tubing with an ultrafiltering cartridge according to Inkinen et al. (2019). Groundwater was pumped from the wells using a submersible pump (Proactive SS-Monsoon XL, GWM Engineering, Kuopio, Finland) to a 20-l plastic canister, sterilized with 70% EtOH and flushed with the sampling fluid, from where the water was pumped through the DEUF capsule cartridge with a peristaltic pump. The flow-through was measured using a water meter and a 10-l bucket. DEUF-capsules were kept cool and sent to further handling in a cool box with ice packs to the laboratory of water microbiology of the Finnish Institute for Health and Welfare, Kuopio.

Microbial biomass was eluted from the DEUF-capsules in laboratory as described in Inkinen et al. (2019). The secondary concentration of DEUF-eluates (100 - 250 mL, corresponding to sample volume of 11 - 80 L) was conducted by filtration through Millipore Express PLUS membrane filters (pore size 0.22 μm, Merck KGaA. Darmstadt, Germany) (Kauppinen et al., 2019) and stored in -75 °C before nucleic acid extraction. Total nucleic acids were extracted from the Express PLUS filters using Chemagic DNA Plant kit (Perkin-Elmer, Waltham, USA) and Kingfisher device (Thermo Fisher Scientific, Waltham, MA, USA) and DNA concentration was measured with a Qubit mini fluorometer using Qubit dsDNA HS assay (Thermo Fisher Scientific, Waltham, MA, USA) as described in Inkinen et al. (2019). Negative control samples of DEUF elution and DNA extraction were also included in the analysis. MiSeq amplicon sequencing of the 16S rRNA gene of bacteria and archaea, and ITS1 region for fungi using 2x300bp paired end protocol was done in commercial laboratory (Eurofins Genomics Ltd., Konstanz, Germany). The primers used were 341F 5'CCTACGGGNGGCWGCAG-3' (Herlemann et al., 2011) and 926R 5'-CCGYCAATTYMTTTRAGTTT-3' (Quince et al., 2011) (V3-V5) for bacteria, 340F 5'-CCCTAYGGGGYGCASCAG-3'(Gantner et al., 2011) and 806R 5'-GGACTACNVGGGTWTCTAAT-3 (Apprill et al., 2015) (V3-V4) for archaea, and 5'-GGAAGTAAAAGTCGTAACAAGG-3' and 5'-GCTGCGTTCTTCATCGATGC-3' for fungal ITS1 (White et al., 1990). Negative controls did not produce amplicons.

Sequence reads were demultiplexed and primer sequences removed before analysis applying the DADA2 (v. 1.30.0) pipeline in Rstudio using R version 4.3.2 (Callahan et al., 2016; R Core Team, 2024) for bacterial and archaeal amplicon sequence variant (ASV) detection. When inspecting the sequence reads, reverse reads of both archaea and bacteria were of poor quality, generating issues in trimming and eventually merging the reads. Thus, the analysis was continued with just forward reads. Reads were filtered using the following parameters (TruncLen=240, maxN=0, maxEE=2, truncQ=2). This removed approximately 4% of reads in each sample. Reads were dereplicated, sequence variants inferred from unique sequences from each sample, chimeras removed and taxonomy assigned using Silva v. 138 taxonomy database (silva\_nr99\_v138.1\_train\_set.fa) (Quast et al., 2013; Yilmaz et al., 2014) for archaea and bacteria.

Fungal sequences were analysed using the ITS-specific variation of version 1.8 of the DADA2 workflow (Callahan et al., 2016). First, ITS primer sequences were detected, primer orientation was verified, and primers were cut from the sequences

using Cutadapt tool (Martin, 2011). After quality trimming with parameters: maxN = 0, maxEE = c(2, 2), truncQ = 2, minLen = 50, rm.phix = TRUE, sequence variants were inferred from unique sequences from each sample, forward and reverse reads were merged, chimeras removed and taxonomy assigned using UNITE general FASTA release (sh general release dynamic 04.04.2024.fasta) (Abarenkov et al., 2024) for fungi.

DADA2-treated sequence data were imported to phyloseq (v.1.46.0) (McMurdie & Holmes, 2013) for data visualisation and statistical analyses in RStudio. Raw sequence data is deposited in NCBI's SRA under BioProject PRJNA1270735.

# 3 Results






# 3.1 Field observations and hydrogeochemistry

Field parameters and the calculated partial pressure of CO<sub>2</sub> (pCO<sub>2</sub>) are presented in Table 2. The groundwater from the sediments had an electrical conductivity ranging from 90 to 204 µS/cm and a pH between 6.2 and 6.8, whereas the water from the bedrock borehole was more mineralized (EC=296 µS/cm) and had a higher pH of 8.4. Temperature range was 5.7 -6.9 °C, and highest dissolved oxygen concentrations were measured at KUU19 and HÄJY11. The field measurements clustered the samples to three groups, R56 clustering clearly separate from the other sites. Oxidation-reduction potential varied significantly between the different samples from lowest in bedrock groundwater R56 (-254 mV) to highest in HÄJY11 and KUU19, corresponding to high dissolved oxygen concentrations in these samples. Chloride concentrations measured in the field shows highest in MIHP15, but this is likely an error as the sensor was not calibrated before this measurement. The laboratory measurement of Cl<sup>-</sup> for MIHP15 was 2.6 mg/l (Supplementary Table 1). Second highest Cl<sup>-</sup> concentration was measured from the bedrock groundwater (R56). Alkalinity varied from 23.5 mg/l in KUU19 to 152 mg/l in R56. The partial pressure of CO<sub>2</sub> (pCO<sub>2</sub>) offers information on the level of system "openness" in relation to soil CO<sub>2</sub>. As water infiltrates the subsurface, it first passes through the unsaturated soil zone, where it equilibrates with carbon dioxide produced by the decomposition of organic matter and root respiration. This results in pCO<sub>2</sub> levels that are higher than those in the atmosphere (Freeze and Cherry, 1979; Appelo and Postma, 2004). In open system conditions, water is in equilibrium with the soil zone CO2 and pCO2 remains constant as carbonate minerals dissolve until equilibrium is reached. Conversely, in a closed system where there is no additional subsurface source of CO<sub>2</sub> available, pCO<sub>2</sub> will gradually decrease as pH increases. Groundwater with a log pCO<sub>2</sub> over -2.0 indicates an open CO<sub>2</sub> system (Clark 2015). The partial pressure of CO<sub>2</sub> (pCO<sub>2</sub>) was calculated based on temperature, pH and HCO<sub>3</sub>, and in all groundwater samples representative of sediment waters, log pCO<sub>2</sub> values ranged between -1.4 and -2.2 atm while the lowest log pCO<sub>2</sub> value in the dataset (-3.3) was observed in bedrock groundwater, suggesting a closed system conditions.

Table 2. Field measurement data of pH, electrical conductivity (EC), temperature (T), dissolved oxygen (DO), oxygen reduction potential (ORP), alkalinity (Alk), chloride concentration (Cl-) from the sampling sites and calculated pCO2. Clustering is based on the field measurements in bold.



| Sample ID | рH  | EC    | T   | DO   | ORP    | Cl⁻    | Alk. (as HCO <sup>3</sup> ) | log pCO <sub>2</sub> |
|-----------|-----|-------|-----|------|--------|--------|-----------------------------|----------------------|
|           |     | μS/cm | °C  | mg/l | mV     | mg/l   | mg/l                        | atm                  |
| MIETO17   | 6,3 | 124,4 | 6,4 | 3,5  | 17,9   | 2,9    | 68,3                        | -1,6                 |
| МІНР6     | 6,8 | 118,0 | 5,9 | 3,3  | 50,5   | 2,2    | 48,8                        | -2,2                 |
| — НÄЈҰ30  | 6,7 | 204,5 | 6,3 | 4,4  | -52,3  | 6,6    | 115,9                       | -1,7                 |
| KUU19     | 6,4 | 89,9  | 6,2 | 47,9 | 104,4  | 4,1    | 23,5                        | -2,2                 |
| LОНІЗО    | 6,6 | 165,2 | 6,4 | 2,7  | 96,0   | 4,6    | 112,2                       | -1,6                 |
| HARJA10   | 6,5 | 178,8 | 5,7 | 3,4  | -91,6  | 7,1    | 92,7                        | -1,7                 |
| MIHP15    | 6,6 | 162,7 | 6,8 | 3,9  | -50,9  | 306,0* | 83,0                        | -1,8                 |
| НÄЈҮ11    | 6,7 | 97,2  | 6,2 | 26,6 | 109.2  | 1,3    | 37,8                        | -2,2                 |
| NOPPA15   | 6,2 | 188,8 | 6,1 | 2,7  | -0,2   | 10,8   | 78,1                        | -1,4                 |
| R56       | 8,4 | 296,2 | 6,9 | 2,7  | -254,0 | 26,6   | 152,5                       | -3,3                 |

\* Chloride sensor was not calibrated before this measurement; therefore, the result may be misleading. Laboratory measurements of chloride concentrations are reported in Supplementary Table 1.

The major ions in groundwaters were Na (46.5mg/l in R56 to 4.82 mg/l in HARJA10), Cl (25 mg/l in R56 to 1.7 mg/l in MIHP6), and Fe (31.5 mg/l in HARJA10, below the detection limit of 0.03 mg/l in HÄJY11 and KUU19 and 0.08 mg/l in MIHP6), and Ca, with highest concentrations in NOPPA15 and HÄJY30 (14.8 mg/l and 15.8 mg/l, respectively) (Supplementary Table 1). Highest sulfur and sulfate concentrations were detected from NOPPA15, and highest organic carbon concentrations were in HARJA10 groundwater (Supplementary Table 1).

Groundwater types were defined and represented on a Piper diagram (Fig. 2) which shows the proportion of major ions (Na<sup>+</sup>, Ca<sup>2+</sup>, Mg<sup>2+</sup>, K<sup>+</sup>, HCO<sub>3</sub><sup>-</sup>, Cl<sup>-</sup>, SO<sub>4</sub><sup>2-</sup>) (Piper 1944). The Diagrams software (Simler, 2012) was used to create the Piper diagram. Groundwater samples from the sediments are of calcium-bicarbonate (Ca-HCO<sub>3</sub>) type or mixed cations-HCO<sub>3</sub>, whereas groundwater collected from the bedrock borehole (R56) beneath the Paloluoma valley is classified as sodium-bicarbonate (Na-

HCO<sub>3</sub>) type, reflecting a more evolved groundwater. NOPPA15 has more chloride than the other samples and KUU19 has some nitrates (4.6 mg/L) (Supplementary Table 1).

Figure 2. Piper diagram for the 10 groundwater samples collected in the study area.

# 280 3.2 Isotopic compositions of hydrogen and oxygen

The variation in the isotopic compositions of oxygen and hydrogen in the analysed water samples was small. The  $\delta^2H$  values range from -91 to -88.2 and the  $\delta^{18}O$  from -12.63 to -12.22. In the Figure 3 the weighted annual average from the south-western coastal site of Olkiluoto and the eastern Finnish city of Kuopio (GNIP database, IAEA), with similar latitude to the study site, are marked on the meteoric water line. The d-excess values Eq. (1), calculated from  $\delta^2H$  and  $\delta^{18}O$  values, range from 9.6 to

10.8, which indicates that the sampled waters are unevaporated and of meteoric origin. This is reinforced by the fact that the samples all plot on or near the meteoric water lines (Table 3, Fig. 3).

$$d - excess = \delta^2 \mathbf{H} - \mathbf{8} * \delta^{18} \mathbf{0} \tag{1}$$

Table 3. The isotopic analysis results and the calculated d-excess values together with the concentrations of strontium and sulphur in the water samples.

| ID       | $\delta^2 H$ (‰) | $\delta^{18}O~(\%)$ | d-excess | <sup>87</sup> Sr/ <sup>86</sup> Sr | $Sr(\mu g/l)$ | $\delta^{34}S$ (‰) | S(mg/l) |
|----------|------------------|---------------------|----------|------------------------------------|---------------|--------------------|---------|
| NOPPA 15 | -89.0            | -12.42              | 10.4     | 0.74535                            | 101           | 1.1                | 5.87    |
| MIHP6    | -91.0            | -12.59              | 9.7      | 0.74029                            | 46.7          | 8.0                | 0.7     |
| MIHP15   | -90.2            | -12.63              | 10.8     | 0.72133                            | 82.7          | 12.6               | 0.34    |
| R56      | -89.4            | -12.51              | 10.7     | 0.73225                            | 91.6          | 23.8               | 0.5     |
| LOHI30   | -89.7            | -12.49              | 10.2     | 0.74159                            | 86            | 30.3               | 0.26    |
| KUU19    | -90.8            | -12.56              | 9.7      | 0.75237                            | 38.3          | 5.9                | 0.39    |
| НÄЈҰ30   | -89.1            | -12.42              | 10.3     | 0.72832                            | 107           | 14.1               | 0.57    |
| MIETO17  | -89.5            | -12.46              | 10.2     | 0.73679                            | 70.8          | 5.3                | 0.93    |
| HARJA10  | -89.9            | -12.49              | 10.0     | 0.73713                            | 74            | 18.5               | 

Figure 3. The isotopic compositions of oxygen and hydrogen from the water samples against the GMWL (Global Meteoric Water line) in red dashed line and the LMWL (Local Meteoric Water Line) in black. Olkiluoto weighted 8-year annual mean (https://inis.iaea.org/collection/NCLCollectionStore/\_Public/46/095/46095041.pdf) and Kuopio GNIP, Global Network of Isotopes in Precipitation (https://www.iaea.org/services/networks/gnip) as references.

# 3.3 Isotopic compositions of Sr and S

The isotopic compositions of strontium and sulphur in the water samples vary between 0.72133 - 0.75237 and 1.1 - 30.3 % respectively (Table 3). The corresponding concentrations of Sr and S in the water samples vary between  $38.3 - 107 \mu g/l$  and  $0.1 - 5.9 \mu g/l$  respectively (Table 3, and Supplementary Table 1). Both elements show noticeable variation (Fig. 4 and 5).


Figure 4. Isotopic composition of sulphur in water samples against the concentrations of sulphur.

Figure 5. Isotopic compositions of strontium against the corresponding reciprocal concentration found in the water samples.

# 3.4 Groundwater residence time indicators

The results of the tritium, helium isotope, neon, SF<sub>6</sub> and CFCs analyses are listed in Table 4.

# 310 **3.4.1 Tritium**

Samples NOPPA15, KUU19, HARJA10 and HÄJY11 had tritium concentrations ranging from 1.6 to 4.8 TU, while samples MIHP6, MIETO17, MIHP15, R56, LOHI30 and HÄJY30, were tritium-free (Table 2). Assuming no mixing, tritium concentrations between 2 and 4.8 TU (NOPPA15, KUU19, HARJA10) indicate a modern component (15 to 40 years old). Tritium concentration below 2 TU (HÄJY11) suggests the presence of a modern component with significant dilution with




older water. The absence of tritium (<0.8 TU) indicates sub-modern waters with ages exceeding 60 years (and the absence of a modern component in this "old" water).

Table 4. Summary of measured <sup>3</sup>H concentrations, calculated tritiogenic Helium 3 (<sup>3</sup>He<sub>trit</sub>), <sup>3</sup>He/<sup>4</sup>He ratio, calculated terrigenic Helium 4 (<sup>4</sup>He<sub>ter</sub>) and atmospheric mixing ratio of CFCs and SF<sub>6</sub>. Text in italics indicates values above solubility equilibrium with modern atmosphere. Tritiogenic Helium 3 was determined with an altitude recharge of 150 m and a recharge temperature of 5°C.

| Sample  | $^{3}H$       | $^{3}$ He <sub>trit</sub> | $^{3}$ He/ $^{4}$ He | <sup>4</sup> He | $^{4}$ He <sub>ter</sub> | Ne       | Ne/He | SF6          | CFC11           | CFC12           | CFC113        |
|---------|---------------|---------------------------|----------------------|-----------------|--------------------------|----------|-------|--------------|-----------------|-----------------|---------------|
|         | TU            | TU                        | ccSTP/g              | ccSTP/g         | ccSTP/g                  | ccSTP/g  |       | fmol/L       | pmol/L          | pmol/L          | pmol/L        |
| NOPPA15 | $4.8 \pm 0.6$ | 87.7±1.14                 | 3.62E-07             | 1.03E-06        | 9.31E-07                 | 3.90E-07 | 0.38  | $18 \pm 4$   | 13 ± 3          | $1.4\pm0.1$     | $0.01\pm0.05$ |
| MIHP6   | < 0.6         | -                         | 1.61E-07             | 1.89E-06        | 1.65E-06                 | 8.76E-07 | 0.46  | 2700         | $0.05\pm0.05$   | $0.02 \pm 0.05$ | < 0.01        |
| MIHP15  | $0.1\pm0.6$   | -                         | 4.08E-08             | 7.25E-06        | 7.12E-06                 | 5.06E-07 | 0.07  | $75\pm23$    | $0.03\pm0.05$   | $0.02\pm0.05$   | < 0.01        |
| R56     | $0.1\pm0.6$   | $188.2 \pm 4.12$          | 2.77E-08             | 1.02E-04        | 1.02E-04                 | 8.56E-07 | 0.01  | $27 \pm 6$   | $0.02\pm0.05$   | $0.01\pm0.05$   | < 0.01        |
| LOHI30  | $0.4\pm0.5$   | -                         | 1.3E-07              | 1.86E-06        | 1.68E-06                 | 6.62E-07 | 0.36  | 990          | $0.08 \pm 0.05$ | $0.03\pm0.05$   | < 0.01        |
| KUU19   | $3.8 \pm 0.7$ | $106.8 \pm 1.33$          | 9.4E-07              | 4.55E-07        | 3.42E-07                 | 4.38E-07 | 0.96  | $110\pm33$   | 230             | $2.1\pm0.2$     | $0.01\pm0.05$ |
| HÄJY30  | $0.2\pm0.4$   | -                         | 6.59E-08             | 3.70E-06        | 3.52E-06                 | 6.88E-07 | 0.19  | 2200         | $0.5\pm0.1$     | $0.02\pm0.05$   | < 0.01        |
| MIETO17 | < 0.6         | -                         | 7.08E-08             | 3.51E-06        | 3.34E-06                 | 6.28E-07 | 0.18  | $100 \pm 31$ | $0.19 \pm 0.05$ | $0.02 \pm 0.05$ | < 0.01        |
| HARJA10 | $4.6 \pm 0.6$ | $60.5\pm0.73$             | 1.36E-06             | 1.71E-07        | 1.13E-07                 | 2.49E-07 | 1.46  | $140 \pm 42$ | $0.07 \pm 0.05$ | $0.01\pm0.05$   | < 0.01        |
| HÄJY11  | $1.6\pm0.7$   | $40.9 \pm 1.56$           | 2.09E-07             | 1.98E-06        | 1.78E-06                 | 7.42E-07 | 0.38  | $32 \pm 7$   | $0.17 \pm 0.05$ | $0.09 \pm 0.05$ | < 0.01        |

# 3.4. 2 Chlorofluorocarbons (CFCs) and sulphur hexafluoride (SF<sub>6</sub>)

All samples containing <sup>3</sup>H also have CFC-11 or CFC-12, confirming the presence of a modern component (Table 2). One exception is HARJA10, which exhibits the presence of <sup>3</sup>H but no CFCs, possibly due to microbial degradation, as this site exhibits anoxic conditions and high bacterial diversity. The only two points with detectable CFC-12 (KUU19 and NOPPA15) are also contaminated with CFC-11. Although the NOPPA15 site partly shares the same recharge area as the other points in the study area, it is located north of the Paloluoma valley and represents a different, shallower system of a modern esker. Tritium-free samples are generally free from CFCs, except for HÄJY30 and MIETO17 which contain some detectable CFC-11. Thus, MIHP6, MIHP15, LOHI30, and the bedrock borehole R56 can be classified as sub-modern (recharged before 1952) due to the absence of CFCs and <sup>3</sup>H. Due to the absence of CFCs or extreme values (contamination), it is not possible to use CFCs for groundwater dating. Therefore, CFCs can only be used here in a qualitative way.

All samples showed SF<sub>6</sub> contents well above solubility equilibrium with modern atmosphere (range was 18-2700 fmol/l) indicating that the excessive groundwater concentrations derive from a local geogenic source. The relative SF<sub>6</sub> contamination







levels might still provide some information on the residence time of groundwater as it appears that samples containing tritium also exhibit relatively lower SF<sub>6</sub> contamination values, apart from the R56 point.

# 3.4.3 Noble gases and <sup>3</sup>H/<sup>3</sup>He dating

All samples show high  ${}^4$ He concentrations, ranging from  $4.55 \times 10^{-7}$  to  $1.0 \times 10^{-4}$  ccSTP/g, which is equivalent to 10-2200 times the solubility equilibrium, indicating a significant source of terrigenic He. The  ${}^3$ He/ ${}^4$ He ratios in tritium-free samples range from  $3 \times 10^{-8}$  to  $7 \times 10^{-8}$ , which are significantly lower than the ratios typical of atmospheric or mantle He. This indicates that the terrigenic He component is entirely radiogenic. The presence of high radiogenic Helium 4 ( ${}^4$ He<sub>rad</sub>) indicates the presence of old groundwater (several thousand years old). The only bedrock borehole R56 has the highest radiogenic helium contents of all the wells. The Ne/He ratios range from 0.01 to 1.46, much lower than the values for dissolved air (4.03).

#### 3.5 Microbial communities

Bacterial 16S rRNA amplicon sequencing was successful from HARJA10, HÄJY30, LOHI30, MIETO17, MIHP15, NOPPA15 and R56. Altogether 996 645 bacterial sequences were retrieved of these, and after quality filtering, denoising and chimera removal steps, 90-94% of the reads were retained in each sample. Altogether 71 different bacterial phyla were detected and ASVs belonging most prevalently to Patescibacteria, Verrucomicrobiota, Chloroflexi and Proteobacteria (Table 5, Supplementary Table 2., 3.). Bacterial communities were diverse and varied from borehole to borehole (Fig. 6a, Supplementary Figure 1a). NOPPA15 hosted the most diverse bacterial community according to Shannon index (H' 6.69), while MIHP15 and bedrock groundwater well R56 exhibited the lowest observed Shannon diversity (H' 3.63 and 2.91, respectively) (Table 5). These communities were more similar with each other compared to other sites. Relatively most abundant bacterial ASVs detected from NOPPA15 affiliated with Gallionella (12% relative abundance), Omnitropha (10%) and several candidate genera of Patescibacteria phyla. The bedrock borehole water R56 hosted a bacterial community with abundant sulfate reducers such as Desulfovibrio (38% relative abundance) in addition to Hydrogenophaga (11%), Ferribacterium (9%) and Acetobacterium (8%). Of these, Hydrogenophaga -affiliating ASVs were either absent or found in very low abundance (

Acidulodesulfobacterales (Tan et al. 2019) (3%). Most common ASVs in LOHI30 were *Candidatus* Omnitrophus (13%), *Ferribacterium* (8%), *Acetobacterium* (8%), Dehalococcoidia (7%), *Rhodoferax* (6%), *Desulfurivibrio* (6%), and *Gallionella* (4%). Similarly, in HARJA10, *Candidatus* Omnitrophus was among the most dominant ASVs (13% relative abundance) together with Desulfobacterota (16%), Sva0485 (10%) and Dehalococcoidia (8%).

Table 5. Bacterial (A), archaeal (B) and fungal (C) diversity in groundwater samples. Observed: number of observed ASVs, Shannon: Shannon H' diversity index, InvSimpson: inversed Simpson diversity index.

A)

| Sample ID | Observed | Shannon | InvSimpson |
|-----------|----------|---------|------------|
| HÄJY30    | 1057     | 3,95    | 6,87       |
| MIETO17   | 1834     | 6,52    | 231,53     |
| NOPPA15   | 2468     | 6,69    | 198,54     |
| R56       | 329      | 2,91    | 6,69       |
| HARJA10   | 1897     | 6,25    | 138,12     |
| MIHP15    | 548      | 3,63    | 13,7       |
| LOHI30    | 3059     | 5,74    | 57,94      |

B)

| Sample ID | Observed | Shannon | InvSimpson |
|-----------|----------|---------|------------|
| НÄЈҮ30    | 1064     | 5,25    | 43,97      |
| MIETO17   | 670      | 4,67    | 20,45      |
| NOPPA15   | 1322     | 5,97    | 155,91     |
| R56       | 287      | 3,81    | 19,12      |
| HARJA10   | 947      | 5,24    | 35,96      |
| MIHP15    | 287      | 2,05    | 3,69       |


C)

| Sample<br>ID | Observed | Shannon | InvSimpson |
|--------------|----------|---------|------------|
| HÄJY30       | 202      | 3,91    | 22,03      |
| MIETO17      | 183      | 3,59    | 15,37      |
| NOPPA15      | 218      | 3,83    | 18,72      |
| R56          | 266      | 4,04    | 18,02      |
| HARJA10      | 261      | 2,66    | 4,80       |



MIHP15 197 2,22 3,05

Archaeal 16S rRNA amplicon sequencing was successful from HARJA10, HÄJY30, MIETO17, MIHP15, NOPPA15 and R56. In total 902 598 archaeal sequences were retrieved, and after quality control and chimera removal steps, 93-96% of the reads were retained (Supplementary Table 2.). A total of 11 different archaeal phyla were identified, and most prevalent phyla were Nanoarchaeota, Micrarchaeota and Crenarchaeota (Supplementary Table 3.). NOPPA15 hosted the most diverse archaeal community (H' 5.97), similar to bacteria (Table 5.). Most archaeal sequences were affiliated with Nanoarchaota phylum in NOPPA15, HÄJY30, MIETO17 and HARJA10, with ASVs belonging to *Woesearcheales* candidate order (74%, 75%, 54%, 40% relative abundances, respectively) (Fig. 6b, Supplementary Figure 1b). Bathyarchaea -affiliating archaeal ASVs were also relatively abundant in these samples. *Candidatus* Methanoperedens and *Methanomassiliicoccales* -affiliating ASVs were abundant in HARJA 10 (18% and 7% relative abundance, respectively), and in R56 (21% and 6%). R56 also had a 11% relative abundance of *Methanobacterium*, in addition to abundant *Woesearchales* (23%) and Bathyarchaeia (20%). MIHP15 hosted the least diverse archaeal community (H' 2.05) that differed from other samples (Table 5). *Methanoregula*, *Methanobacterium*, and *Methanospirillum* -affiliating ASVs were dominating the archaeal community in this borehole (50%, 28% and 12 % relative abundances, respectively).

Fungal ITS1 amplicon sequencing was successful from the same samples as archaea. Altogether 649 988 sequences were retrieved, but after rigorous quality filtering and chimera removal steps, on average only 41% of sequences were kept (Supplementary Table 2.). According to the Shannon index, the most diverse fungal community was observed in R56 (H' 4.04) and least diverse in MIHP15 (H' 2.22) (Table 5.). Most of the detected fungal phyla belonged to either Ascomycota or Basidiomycota, and relatively most abundant ASVs affiliated with Cladosporium and *Bulleribasidiaceae* (Fig. 6c, Supplementary Figure 1c). In addition to these, R56 fungal community was composed of *Claviceps* and *Rhodotorula* (8% and 6% relative abundances). Fungal communities in NOPPA15, HÄJY30, MIHP15 and MIETO17 all had *Filobasidium* - affiliating ASVs (7%, 9%, 5% and 13% of the community, respectively). *Itersonilia* affiliating ASV was more typical to HÄJY30 (9%) than to other samples, and *Melanommataceae* -affiliating ASV was most relatively most abundant in NOPPA15 (7%). A large portion (57%) of the fungal community in HARJA10 could not be identified beyond kingdom-level.

Figure 6. Groundwater microbial communities. Bar plots show the relative abundances of different A) bacterial, B) archaeal, and C) fungal phyla and heatmaps represent 25 most abundant orders of A) bacteria, B) archaea and C) fungi.

The results showed that the groundwater microbial communities are diverse and composed of different populations in different boreholes along the aquifer in Kurikka. The bedrock-related microbial community has some special features compared to the

groundwaters from shallower unconsolidated aquifers. The site at a more superficial esker system with relatively young groundwater hosted the most diverse prokaryotic community, while bedrock with the oldest waters the least diverse.

#### 4 Discussion






Hydrogeological studies rely on use of various tracers, most often field measurement data and laboratory analysis of chemical parameters, i.e. main ion and trace element concentrations. Natural tracers have been expanded to include the use of different isotopes and microbial communities in addition to groundwater age revealing gases (CFCs, SF<sub>6</sub>). These techniques can be used to assess the physical, biological and chemical processes that take place in a hydrogeological setting (Divine and McDonnell 2005).

# 4.1 The origin and age of groundwater in Kurikka according to tested tracers

Geochemistry of the sampled water provided initial information about the origin of groundwater in the complex aquifer system in Kurikka. The distinct EC values between sediment groundwater (90-204 µS/cm) and bedrock groundwater (296 µS/cm) indicate different mineralization levels. Higher EC in bedrock groundwater suggests more evolved water, indicating longer residence time allowing for mineral weathering influencing total dissolved solids (TDS). Similarly, pH differences (6.2-6.8 in sediment groundwater vs. 8.4 in bedrock groundwater) suggest longer interaction time with minerals. In deep bedrock environment in Finland, where fluids have retention times of tens of millions of years, have groundwaters also have high TDS, EC and pH (Kietäväinen et al. 2013). Major ion composition provided similar characterization of groundwater types. Sediment groundwater being calcium-bicarbonate (Ca-HCO<sub>3</sub>) or mixed cations-HCO<sub>3</sub> type, and bedrock groundwater being sodium-bicarbonate (Na-HCO<sub>3</sub>) type, indicate different geochemical processes and sources. Often the presence of nitrates and higher chloride levels can indicate anthropogenic influences, such as agriculture, or recent recharge events in groundwater. However, noticeably higher Cl<sup>-</sup> concentrations from bedrock groundwater are more likely to originate from the long water-rock interaction rather than anthropogenic influence. In our study, only one sample (KUU19) had a nitrate concentration above the detection limit (Supplementary Table 1). This is likely explained by the agricultural activity in the area. Furthermore, KUU19 is likely influenced by intensive test pumping (700-1000 m3/day) from a nearby production well, causing more dynamic groundwater flow compared to natural conditions. This can explain the anomalies seen in the KUU19 sample.

The partial pressure of CO<sub>2</sub> (pCO<sub>2</sub>) helps understand the geochemical conditions of the groundwater and indicates the degrees of "openness" with the soil atmosphere (Clark 2015). In all water samples representative of sediment groundwaters, pCO<sub>2</sub> values higher than the atmospheric values suggest open-system conditions where groundwater is in equilibrium with the "soil zone" CO<sub>2</sub> (Table 2). In contrast, the pCO<sub>2</sub> value in bedrock groundwater suggests closed-system conditions, without CO<sub>2</sub> input from the soil zone.

The  $\delta^2H$  and  $\delta^{18}O$  values from the sampled waters lie on or near the meteoric water lines except for the R56 and MIHP15 samples, that plot, albeit very slightly, above the meteoric water lines. Bedrock groundwater samples plot normally above the







meteoric water line in case they have experienced long-term water-rock interaction (Kietäväinen et al. 2013). The longer residence time enables the abiotic chemical processes in the water-rock interaction, that can fractionate the isotopic compositions of the water stable isotopes (Kietäväinen et al. 2013). These processes may also explain the MIHP15 water sample plotting slightly above the meteoric water lines, since the MIHP15 well reaches the lowermost aquifer, and most likely has active hydraulic connections to the fractured bedrock below. In boreal regions variation on the plots can be seen due to seasonality (Clark 2015). The small variation in the isotopic compositions of hydrogen and oxygen in the groundwater samples indicates similar seasonality to the precipitation that has infiltrated the aquifer system, although here the one-off sampling says little about the seasonal changes. There is no evaporation component from a potential surface water source in the sampled waters.

The local geology is rich in silicate minerals (biotite paragneiss, granite and diorite) that have a relatively high rubidium content. Rubidium <sup>87</sup>Rb decays to <sup>87</sup>Sr, and the enriched <sup>87</sup>Sr/<sup>86</sup>Sr values could imply either felsic mineralogy in the study area or the process of silicate weathering, both of which produce heavier strontium isotopic compositions (Ikonen et al. 2025, Négrel et al. 2018). The isotopic composition of strontium in the groundwater samples varied significantly. Although the bedrock in Kurikka is rich in rubidium-bearing mica (biotite, muscovite) minerals, which produce enriched Sr isotopic values to the bedrock groundwater, the Sr isotopic composition in the R56 bedrock groundwater sample was lower than in most of the unconsolidated aquifer water samples excluding HÄJY30 and MIHP15. A study done in Palmottu, southern Finland with a similar geology to Kurikka found the 87Sr/86Sr values ranging between 0.71999 and 0.75079 in the bedrock groundwater samples (Négrel et al. 2003), which correspond to R56. If there are hydraulic connections, the mixing of different 87Sr/86Sr end-member waters is the most likely explanation for the varying Sr isotopic compositions also in the lower-most unconsolidated aquifer water samples.

As with the isotopic compositions of strontium the variation in the  $\delta^{34}$ S values is noticeable (1.1 – 30.3 %). The microbial process is the most important cause of fractionation for the sulphur isotopes. Due to the microbial reduction of sulphate, the 460 lighter <sup>32</sup>S isotope is removed from the solution (Kaplan and Rittenberg 1964). A clear influence of the sulphate reducing bacteria can be seen in the samples MIHP15, HÄJY30, HARJA10, R56 and LOHI30 (see section 4.2.1), which produce the enriched  $\delta^{34}$ S values ( $\delta^{34}$ S >12.6). The rest of the samples KUU19, MIETO17, MIHP6 and HÄJY11, ( $\delta^{34}$ S values 5.3 – 8.8 ‰), seem to reflect the atmospheric isotopic compositions of sulphur (-5 – 10 ‰) (Clark and Fritz 1997). In addition, the  $\delta^{34}$ S value of NOPPA15 stands out, similarly to  ${}^{87}Sr/{}^{86}Sr$ , in the case of  $\delta^{34}S$  with a lower isotopic composition and higher sulphur concentration to the rest of the samples (Fig. 4).

The oxidation-reduction potential in NOPPA15 was close to 0, so neither oxidation nor reduction reactions are strongly favoured. NOPPA15 had lighter isotopic composition of sulphur compared to other samples, suggesting minimal influence from biological processes that typically fractionate sulphur. Furthermore, different mineralogy and water flow paths from the recharge site also probably play a part in the isotope geochemistry in the sampled water from NOPPA15. Thus, NOPPA15 stands out due to its location reflecting a context of modern esker in the Nenattömänluoma valley (Fig. 1).







Residence time indicators show the presence of a modern water component in several sediment groundwater locations. This is consistent with a less evolved water type and pCO<sub>2</sub> values indicative of open-system conditions. In contrast, the only bedrock borehole R56 represents sub-modern water (free of CFCs and <sup>3</sup>H, with the highest <sup>4</sup>He<sub>rad</sub> concentration), with a more evolved water type, and pCO<sub>2</sub> values reflecting a closed system. The elevated <sup>4</sup>He<sub>rad</sub> concentrations in all samples complicates the evaluation of groundwater residence times and mixing proportions using the <sup>3</sup>H-<sup>3</sup>He technique (Shapiro et al. 1998). When radiogenic helium 4 is abundant, accurately separating the tritiogenic helium component becomes challenging and highly uncertain. This makes both reliable dating and the determination of mixing proportions unattainable here. The use of CFCs as tracers is constrained by their absence, localized contamination (e.g., NOPPA15, KUU19), or microbial degradation in the anoxic conditions of this environment (e.g., HARJA10). Occurrences of extreme SF<sub>6</sub> values due to natural sources have also been reported in southern Sweden (Åkesson et al. 2015). The SF<sub>6</sub> contamination in all samples due to the geogenic sources (presence of granite, granodiorite) also rules out the use of this tracer in the Kurikka aquifer system.

Microbial diversity followed roughly the groundwater residence times. Most diverse bacterial and archaeal communities were detected from sites with a high fraction of young groundwater, while least diverse prokaryotic communities were in bedrock groundwater with old groundwater. Previous study has shown a similar distinction between old and younger groundwater microbial communities (Ben Maamar et al. 2015). This can be a result of nutrient limitation: older groundwaters are considered more oligotrophic and energy-limited than younger groundwaters (Griebler & Lueders, 2009). Thus, older fluids provide a less diverse array of nutrient and energy sources for microbial inhabitants. Consequently, only those microbes capable of utilizing the available resources can flourish in such environments. Of the deep aquifer samples, those falling in Häjynluoma (HÄJY30) or Paloluoma (R56, MIHP15) buried valleys, were less diverse compared to those that were in between these (MIETO17, HARJA10, LOHI30). Although all sampled groundwaters were significantly younger, the diversity indices were approximately at similar level with studies of significantly deeper and older groundwater samples from Fennoscandian bedrock (Kietäväinen et al., 2014; Purkamo et al., 2016). The bedrock groundwater well R56 also hosted a microbial community with a distinct member, *Hydrogenophaga* -affiliating bacteria, which have been detected previously from older groundwaters from Canada (Ruff et al., 2023) and millions of years old, deep drillhole fluids from Fennoscandian bedrock in Finland (e.g., Purkamo et al., 2013, Kietäväinen et al., 2014).

On the other hand, fungal communities were most diverse in the bedrock well (R56) containing old groundwater, while the lowest diversity was observed in the deep unconsolidated aquifer site (MIHP15), also with a significant portion of old groundwater. This suggests that, unlike prokaryotic communities, the diversity of fungal communities is not influenced by the groundwater residence time.

# 4.2 Hydrogeochemistry is reflected in the microbial community structure of the groundwater

#### 4.2.1 Sulphur

Concentrations of sulphur in the water samples were small, except for the upper aquifer water from the observation well NOPPA15 (5.87 mg/l, Fig. 4), but the values for  $\delta^{34}$ S showed notable variation from 1.1 for NOPPA15 to 30.3 for LOHI30.

The enriched end of the isotopic compositions of sulphur are seen in the samples from LOHI30, R56, HARJA10, HÄJY30, and MIHP15, all with significant proportions of sulphate-reducer communities, explaining the enrichment of the δ<sup>34</sup>S values. Other samples also had minor sulphur reducing bacterial populations, the lowest abundance of sulphate reducing bacteria was seen in the water sample from NOPPA15, together with the higher concentrations of sulphur, explaining the low δ<sup>34</sup>S values (Fig. 4). The microbial investigation demonstrated abundant sulphur-reducing bacteria affiliated with *Desulfovibrio* in R56 and MIHP15, and *Desulfosporosinus*-affiliating sulphur reducers in HÄJY30. HARJA10 had several ASVs belonging to Desulfobacterota phylum, and notable relative abundance of Sva0485, recently identified as *Candidatus* Acidulodesulfobacterales, with potential ability to both reduction and oxidation of sulphur species, depending on the environment's oxygen availability (Tan et al., 2019).

#### 515 **4.2.2. Iron**




Iron cycling microbes were detected in in LOHI30, HÄJY30 and MIETO1, in which iron concentrations were between 1,47-9,75 mg/l. Fe (II) oxidizers such as Gallionella were detected in abundance in NOPPA15, LOHI30 and MIETO17, despite the oxygen-depleted and reductive environment. A recent pangenomic study reflected that while traditionally Gallionellaceae have been considered obligate aerobes, some may use other strategies for energy production (Hoover et al., 2023) that can explain the detection of iron oxidizers in these groundwater wells. No iron cycling microbes were detected in HARJA10 despite its highest iron concentrations. This suggests that for example chemolithoautotrophic *Gallionellaceae*-affiliating ASVs, which were abundant at other sites, were outcompeted by other organisms. *Gallionella*-type of bacteria use inorganic carbon for growth, and iron oxidation typically occurs in environments scarce in digestible organic carbon (Hallbeck & Pedersen, 2014). In contrast, HARJA10 had abundant organic carbon, which likely supported a different microbial community.

In addition, iron reducing and strict anaerobes *Ferribacterium* ASVs were detected from LOHI30 and R56. Interestingly, LOHI30 and MIETO17 bacterial communities hosted *Rhodoferax* that has been shown to be capable of both Fe(II) oxidation and Fe(III) reduction depending on the environmental conditions (Kato & Ohkuma, 2021). This kind of bifunctional capacity can provide an advantage for the organism in environments changing from aerobic to anoxic conditions, and lead into a significant role in recycling Fe.

# **4.2.3 Carbon**

A large proportion of microbial communities in Kurikka groundwater were affiliated with heterotrophic fermenters, such as *Omnitropha*. Previously known as candidate phylum OP3, and currently named Omnitrophota, they are a diverse group of microbes divided to seven order-level clades. They are commonly detected in groundwater environments and have been shown to represent significant proportions of the microbial community (Perez-Molphe-Montoya et al., 2022; Suominen et al., 2021; Williams et al., 2021). *Omnitropha* -affiliating bacteria have also been described as a type species from deep underground mine groundwaters (Momper et al., 2017). In addition to their role in mediating carbon cycling, mostly relying on organic carbon fermentation, some Omnitropha are capable of mixotrophy and utilization of the Wood-Ljungdahl pathway to fix





carbon dioxide and producing acetate (Perez-Molphe-Montoya et al., 2022). Interestingly, previous studies show dominant *Omnitropha* or *Verrucomicrobiales* in environments with enriched elevated concentrations of trace metals, such as chromium or strontium (Bärenstrauch et al., 2022; Zhang et al., 2021). A high relative abundance of *Omnitrophus* was observed in our samples, in which Sr concentrations were highest and chromium concentrations above the detection limit.

Another potential acetate-producing microbial group, Acetobacterium, was present in some of the studied sites (HARJA10, LOHI30, MIHP15, R56). Homoacetogenic bacteria like Acetobacterium reduce CO<sub>2</sub> to produce acetate, using hydrogen as electron donor, therefore functioning as primary producers in groundwater (Lever, 2012). Acetate production can support not only heterotrophic microbes but also sulfate reducers by providing a continuous source of suitable substrate for energy and growth.

Similarly to previous studies, Patescibacteria-affiliating ASVs were prevalent in groundwater prokaryotic communities (Lyons et al., 2021; Schwab et al., 2017). Patescibacteria are known for their small cell size, symbiotic lifestyles and streamlined genomes. Their carbon metabolism is likely fermentative, thus Patescibacteria are contributing to breakdown and cycling of organic carbon in groundwater (Luef et al., 2015). The minimal gene array for biosynthetic and metabolic pathways makes them highly dependent on other microbes in the community, but on the other hand, they require minimal resources and are able to efficiently utilize the limited nutrients and energy sources in groundwater (Bärenstrauch et al., 2022; Chaudhari et al., 2021). Patescibacteria are abundant in both oxic and anoxic groundwaters, often co-occurring with Omnitrophota and Nitrospirota, both detected in the Kurikka aquifer system. This may indicate networking and potential dependencies between these microbial components of the community (Chaudhari et al., 2021).

# 4.2.4 Nitrogen

HARJA10, MIETO17 and NOPPA15 communities had abundant populations of Nitrospirae microbes in groundwater.

Cultivated members of Nitrospira are known to be able to perform nitrite oxidation, complete ammonia oxidation in aerobic conditions, or anaerobic iron oxidation (Daims et al., 2015; Rosenberg et al., 2014). Recent studies highlight the importance of Nitrospirae in groundwater ecosystems and their diverse metabolic abilities (Mosley et al., 2024; Poghosyan et al., 2019; Schwab et al., 2017). Nitrospirota lineages differentiate according to oxygen availability (Mosley et al., 2024). In oxic aquifers, Nitrospirota communities differ from those in anoxic aquifers, where for example *Thermodesulfovibrio* are more dominant.

Although oxygen was present in the Kurikka aquifer, we mostly detected these anaerobic Nitrospirota lineages. This aligns with findings of Mosley et al. (2024) which show that the majority of Nitrospirota genomes recovered from aquifers possess metabolic traits linked to anaerobic nitrogen and sulphur transformations.

# 4.3 Usability of tested tracers in complex aquifers

While this study provides preliminary insights into the Kurikka buried valley aquifer system, it is important to acknowledge several limitations. Most observation wells have long screens (12 to 30 m) which can result in samples containing mixtures of water representative of different flow paths. The analyses were performed after a single sampling session. This excludes the








seasonal changes in redox conditions that might affect the S isotopic compositions, or the seasonal changes in the isotopic compositions of water stable isotopes in precipitation. Furthermore, the sampling was done from only 10 locations, making it challenging to determine the endmembers for Sr isotopic compositions for possible mixing evaluations. For a more complete assessment of the end-member compositions more sampling around the study area can be recommended.

Challenges may arise when interpreting the results from isotopic analyses, especially in geologically or hydrogeologically complex surroundings, where various processes cause large variation in the isotopic compositions. This can be assumed in the Kurikka study site. The isotopic compositions of strontium and sulphur tell of local geochemical and biogeochemical complexity, and in the case of sulphur, of the variation in the microbial communities. Mixing of groundwaters in the aquifer system can potentially be determined with the help of these results in the future, as more sampling campaigns are made, the test pumping continues, and further knowledge of the aquifer system is gained.

The groundwater dating methods used in the study area have major constraints. These issues come from CFCs contamination, geological influence impacting SF<sub>6</sub> and radiogenic <sup>4</sup>He, and the absence of these tracers in several sampling sites. Future sampling should prioritize tritium as well as radiocarbon to characterize the old groundwater component in the system.

Using microbial communities as tracers in groundwater studies presents several challenges. While distinct community profiles can be observed across samples, many phyla and orders are shared among all sites, limiting their discriminatory power. Greater resolution might be achieved by examining lower taxonomic levels; however, partial 16S rRNA gene sequencing typically lacks species-level accuracy (Johnson et al. 2019). This can be circumvented by sequencing the whole 16S rRNA gene using long-read approaches, however, these are still more costly and not as well established as the amplicon sequencing methodology. Certain taxa, such as *Hydrogenophaga*, may serve as indicators of specific environments like bedrock groundwater, but detecting them reliably would require targeted methods. Moreover, microbial taxonomy alone does not reveal functional roles, which are essential for interpreting biogeochemical processes.

### Conclusions

This study highlighted the unique characteristics of deep groundwaters found in the buried valley aquifer system in Kurikka, focusing on combined isotopic, residence time and microbial tracer analyses. These foundational aspects provide a comprehensive understanding of the groundwater system. The findings indicate that the groundwater in the Paloluoma Valley and Harjankylä in the Kyrönjoki Valley appear to be younger compared to the other areas in the Kyrönjoki Valley. This is attributed to the proximity of recharge areas, a crucial observation for groundwater management during extensive water extraction periods. Isotopic compositions of oxygen and hydrogen show no surface water component in the groundwater samples, and strontium isotopes provide reference to local mineralogy and a reference for future mixing studies, assuming the pumping tests will influence the hydraulic connections in the aquifer system. Based on the isotope geochemistry done from the sampled waters, it seems there are hydraulic connections between the lower most aquifer and the fractured bedrock below it. As to the extent of the fractured bedrock as a water reservoir, it is difficult to comment based on this study only. Groundwater

residence time estimates from this analysis indicate predominantly modern water. This suggests that if mixing occurs between water from the bedrock surface and the lowermost unconsolidated aquifer, groundwater flow in the fractured bedrock is likely relatively unrestricted.

The isotopic compositions of sulphur in the groundwater samples have a strong link to the microbial communities, which vary between the different sites. Especially the bedrock groundwater microbial community feature specific organisms that were not detected from the shallow aquifer groundwaters. The prokaryotic community diversity was higher in younger groundwaters and low in older bedrock-originating groundwater.

It is important to note that further research is needed to explore various aspects, such as the origin of the water and the connections between different aquifers in the valley. The insights gained here are necessary for a more detailed understanding and effective management of groundwater resources. Furthermore, they provide a reference for various parameters for future studies to reflect and rely upon, prior to large-scale pumping tests and the planned groundwater extraction from the aquifer system.

#### 620 Author contributions


LP, JI and MAP jointly conceptualized the idea, conducted the investigation, analysed the data and wrote and edited the original manuscript. NP took part in conceptualization, provided funding and resources and commented on the original manuscript. MM took part in investigation and commented on the original manuscript. AH took part in conceptualization, investigation and analysis of the data and writing and editing of the original manuscript. TP provided resources and participated in conceptualization of the idea. IM provided funding and resources and mentoring during idea conceptualization.

The authors declare that they have no conflict of interest.

# Acknowledgements

Arto Pullinen, Salla Valpola, Vaula Lukkarinen and Kim Wennman are thanked for their help during the fieldwork. Yann Lahaye, Mia Tiljander, Meiru Zhou and Katariina Issukka at the GTK laboratory, and the personnel of laboratory of water microbiology at THL are acknowledged for their contribution to laboratory analyses. Nina Hendriksson and Aleksi Tuunainen are thanked for the helpful comments and aid with the characteristics of groundwater wells. Authors also thank the Kurikka Aquifer project consortium partners: Kurikan Vesihuolto Oy, Vaasan Vesi, and ELY Centre for South Ostrobothnia (EPOELY) for funding and resources for the project. We are also sincerely thankful for all private landowners allowing access to the sampling sites. During the preparation of this manuscript the authors used Microsoft Copilot to improve language and readability. After using this tool, the authors reviewed and edited the content as needed and take full responsibility for the content of the publication.

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
