# Peer review of "Microbial communities and isotopes as novel tracers for groundwater flow paths in the multi-layered aquifer system in Kurikka, western Finland"

_EGUsphere, 2025_

## Referee Comment (RC2)

**Reviewer Report**

**Summary**

The manuscript presents a multi-tracer investigation of a buried-valley aquifer system in Kurikka (Finland), combining hydrochemistry, environmental isotopes, noble-gas-based age tracers, and microbial community analyses across several wells. The overall aim appears to be a characterization of the multi-layered aquifer using complementary methods (i.e., combining tracer hydrology and microbiology).

Recent studies have already demonstrated the potential of combining microbiological and environmental tracers to examine groundwater systems. Situating the study more explicitly within this existing frame of work would help contextualize the approach and clarify its contribution.

**Major Concerns**

**1. Lack of research question, hypothesis, and conceptual focus**

The manuscript does not articulate a research question or hypothesis, nor does it explain why this specific suite of tracers and microbiological approaches is required to address a defined problem. As a result, the study reads primarily as a catalogue of measurements, and the discussion remains largely descriptive. Without a defined scientific objective, it is difficult to evaluate the relevance or interpretation of the results.

**2. Insufficient treatment and interpretation of noble gas, ${\rm CFC/SF_6},$ and $^3{\rm H}/^3{\rm He}$ data**

The tracer dataset contains internal inconsistencies that remain unexplained:

- Ne concentrations exceed air-saturated water (ASW) values by 121–427%, which is far above typical excess-air ranges expected for such aquifers.
- Some CFC and SF6 values also exceed ASW by large margins.
- Dissolved O2 values reach several hundred percent of ASW (23–401%), including values incompatible with waters sampled at depths up to 160 m.
- Tritium values are inconsistently reported, and negative 3He\* values appear to have been omitted without comment. When applying the unfractionated air (UA) model to the published data, similar 3H/3He results can be reproduced; however, this approach also yields negative 3He\* values, which seem to have been excluded from the table without explanation.
- There is no indication how terrigenic and tritiogenic helium concentrations were calculated, which makes them difficult to be interpreted.

Such large oversaturations in  $O_2$ , Ne, CFC, and SF6 could stem from sampling artefacts, air contamination, or analytical issues. If real, they require detailed mechanistic discussion. As presented, the tracer dataset cannot be reliably interpreted.

**3. Lack of methodological transparency**

Several essential methodological elements are missing, preventing reproducibility. Examples include:

- There is no indication of how terrigenic and tritiogenic helium concentrations were converted into age, which makes their interpretation difficult.
- No description of how terrigenic 4He was separated from atmospheric components.
- No justification for omitting negative 3He\*-derived ages.
- Missing operational details such as pumping setup, flow rates, sample handling, and field conditions.

These elements are critical for evaluating reliability, and the present level of detail is insufficient.

**4. Insufficient integration between microbiological and tracer datasets**

The manuscript reports microbial and hydrochemical results but does not clearly connect them. The potential value of combining microbial community structure with isotopic and noble gas tracers is not demonstrated. The discussion does not link redox conditions, residence times, mixing processes, or flow systems to microbial patterns. As a result, the combined dataset does not produce a unified hydrogeological interpretation.

**5. Limited site characterization**

The hydrogeological context of the study area is not described in sufficient detail. Flow directions, stratigraphy, connectivity between aquifers, and the position of the unconsolidated aquifer relative to the sampled wells are not clearly presented. This makes it difficult to interpret the tracer results or understand how the sampling locations relate to flow paths.

**6. Figures and tables require substantial revision**

Several figures and tables are not sufficiently clear, contain redundant information, or do not contribute meaningfully to the interpretation. A reduction in the number of figures, focusing on more integrative and hydrologically relevant visualizations, would strengthen the manuscript.

Suggested improvements include:

- Combined tracer—isotope heatmaps linked with microbial community data.
- A 3He/4He vs. Ne/4He diagram including ASW, air, crustal, and other endmembers.
- $\bullet$  Clearer representation of a quifer geometry and flow systems.

There is also interpretive potential in several existing figures. For example, the patterns in the current Figure 5 showing the strontium isotopes data may indicate mixing trends or groundwater evolution relevant to groundwater flow.

Table 4 contains missing uncertainties, inconsistent reporting, and unexplained omissions. If replicates were taken, standard deviations should be reported consistently.

**7. Interpretation remains descriptive**

Without a guiding question or conceptual framework, the manuscript does not progress beyond description. A hydrogeological model relating the measured variables to flow, recharge, mixing, or redox evolution would greatly improve interpretive strength. Some figures and questions are added here which might stimulate some interesting scientific discussion.

**Additional suggestion:** Given the complexity of noble gas and age-tracer interpretation, and the inconsistencies identified in the dataset, it may be beneficial for the authors to consult with a tracer hydrology specialist to ensure that sampling, modelling, and interpretation follow established practices.

**Recommendation**

Given the absence of a research question, inconsistencies in key tracer datasets, limited methodological transparency, and lack of integrated interpretation, I cannot recommend the manuscript for publication in its current form. Addressing these concerns would require major restructuring, re-analysis of the tracer datasets, improved methodological documentation, and development of a focused conceptual framework.

**Figures**

Figure 1: Illustrative example of a  ${}^{3}\text{He}/{}^{4}\text{He}$  vs. Ne/ ${}^{4}\text{He}$  plot that could support interpretation of water origins. Provided solely to clarify the suggestion in the review.

Figure 2: Illustrative reproduction of the manuscript's Figure 5 to highlight possible mixing relationships. Provided for review purposes only.

Figure 3: Correlations between excess Ne and various microbial metrics highlight additional questions arising from the dataset. Although the groundwater should be anoxic at the reported depths, the samples contain substantial dissolved oxygen, while the archaeal and bacterial communities are characteristic of strictly anoxic environments. These contrasting patterns raise the possibility of air contamination. The apparent association between excess Ne and fungal diversity is also notable. While anaerobic fungi are known from certain specialised environments, their occurrence remains uncommon. Such observations therefore warrant careful interpretation in light of the unusually high gas oversaturation reported.

---

## Referee Comment (RC3)

The paper by Purkamo *et al.* is an extensive study of deep granite groundwater in Kurikka, Finland. The study is notable for its analysis of a large number of hydrochemical, geochemical and biological parameters, which distinguishes it from comparable investigations. This comprehensive data collection has the potential to provide valuable insights into the system.

However, the manuscript appears to be in the early stages of development and could benefit from streamlining and more concise data representation. While the data is valuable for characterizing subsurface ecosystems, I would advise against publishing it in its current form.

Major comments:

1) The study contains a wealth of data from a variety of locations, but several aspects of the data analysis and presentation require substantial improvement to enable the reader to follow the relationship between sites and measurements and convey key messages. We recommend that the authors perform statistical analysis to link their data and identify important trends and distinguishing/similar features between sites. Furthermore, Tables and Figures should be improved for readability and clarity.

2) The central narrative of the manuscript requires reconsideration. The title and conclusions regarding subsurface connectivity and transport processes are not adequately supported by the data because no direct measurements or modeling of groundwater flow paths, hydraulic connectivity between boreholes, or microbial advection were performed. The study is based on single-time-point sampling without replication which is insufficient to infer connectivity or transport between sites.

3) The general writing of the manuscript should be improved and the introduction and discussion require further integration. In the current form many sections are disconnected and include unnecessary details. Many items are introduced but the connection between topics and the rational for inclusion is missing.

Minor comments:

Title

There is no modeling or measurement of flow path in the paper. Its rather the microbial and hydrogeological characterization of the multi-layered aquifer system in Kurikka, Western Finland.

Figures and Tables

**Figure 1:** Colors of the cross section and colors of the legend do not match.

**Table 1:** Please designate if MM.DD.YYYY or DD.MM.YYYY

**Figure 4 and 5:** Please remove border around the figures. Missing units on the x axis.

Introduction

**Line 69:** Biological isotope fractionation is a passive process. The slight difference in mass influences e.g. bond length and vibrational entropy within the molecule, making the enzymes work more

efficiently with the 'more flexible' light isotope. It's not an active 'decision' of the microbe. Please rephrase.

**Line 81:** Not all groundwater systems are "cool." Also, "cool" is a subjective description of temperature. Please rephrase.

**Line 94:** This does not make sense. In order to survive, many groundwater microbes must reduce oxygen, nitrate, sulfate, or iron and oxidize organic carbon (in the case of heterotrophs) or reduced sulfur, nitrogen, or iron to produce energy for $CO_2$ fixation (in the case of chemolithoautotrophs).

**Line 106:** Please shortly introduce how microbial communities might be of interest for contaminant remediation. This is the first time it is named as an aim of the study.

**Line 108:** 'The region' or 'This region'

Methods

**Line 128:** Adding sub-headers to the Materials and Methods section would greatly improve clarity.

**Line 137:** Would be good if the weather conditions during this sampling were put into a broader context, e.g. did the precipitation differ significantly from the annual average, how variable are temperature and precipitation over a year?

**Lines 224 to 225:** Please provide additional information on visualization and statistical methods.

Results

**Line 235:** If the chloride measurement is wrong, it can be excluded.

**Line 239:** "Level of system openness" sounds strange. Connection to surface, maybe? Please rephrase.

**Line 240ff:** This doesn't seem to belong to the results. Introduction?

**Line 245f:** Calculations can be explained in the methods section.

**Line 282:** Units are missing.

**Line 284:** Units are missing.

**Line 309:** Section 3.4 is only one sentence. If this data is not relevant for the main storyline it can go to the supplement.

**Line 322:** If there are exceptions, it's not "all samples."

**Line 232:** What is that dissolved oxygen concentration?

**Lines 238 to 245:** This information should be moved to methods.

**Line 344:** It is unclear how the read numbers are distributed across the samples.

**Line 345:** Table 5 contains diversity indices not the microbial community data.

**Line 386:** This is surprisingly little and it might be good to revisit the bioinformatics pipeline to determine why so little was retained. Chimera removal can lead to relatively large losses for fungal sequences, which might have happened here but this should be clarified.

Discussion

The discussion of fungal and archaeal taxa beyond the biodiversity patterns is missing. A short discussion of their potential role in the studied system would be important.

**Line 422:** Please adjust typo: "…have groundwaters also have high TDS…"

**Line 429:** This is an interesting connection to above ground activity, and it would be helpful to have better contextualization (potentially in the introduction together with the bioremediation of aquifer contaminants).

**Lines 432 to 436:** This would benefit from better integration into the other data and seems detached from the rest of the discussion.

**Lines 459 and 460:** Please define the "microbial process".

**Lines 460 to 466:** This would be an interesting place for an introduction of which microbes in the microbial community could be performing sulphate reduction.

**Lines 507 to 509:** Please expand on the reasons for low Sulfate reducers in the high sulfur well. Also please clearly define which taxa are considered sulfate reducing. This is also true for the iron and nitrogen cycling microbes.

**Line 519ff:** If dissolved oxygen is present at all sites, aerobes should be able to grow.

**Line 546:** Interesting, please clarify and cite literature.

**Lines 565 to 567:** Would be an interesting place to expand on the role of nitrogen cycling microbes in contaminant remediation.

**Line 570:** What about a correlation analysis between microbial species and environmental parameters?

**Line 615:** In what way are the results relevant for groundwater extraction endeavors? This could be discussed a bit more.

---

## Author Comment (AC5)

**Comments on Reviewer2**

1. Lack of research question, hypothesis, and conceptual focus

We have revised the manuscript to clearly articulate the larger goal of the research initiative that this study is part of. Specifically, we now state that the study aims to determine the natural state of the aquifer, providing the baseline hydrogeochemical and microbiological characteristics of the aquifer system prior to large-scale water abstraction.

We have clarified that the research question focuses on whether microbial community structure, i.e. "microbial fingerprint", combined with selected chemical and isotopic tracers, can provide insights into groundwater origin, age, and connectivity within a complex buried valley aquifer system. The hypothesis is that microbial community structures, when integrated with multitracer hydrogeochemical approaches, offer a novel tool for characterizing flow dynamics and aquifer connectivity. These additions explain why this specific suite of tracers and microbiological approaches was selected: to evaluate their applicability and compatibility for aquifer characterization and to support sustainable groundwater management. We believe these revisions address the concern that the study previously appeared as a catalogue of measurements and strengthen the interpretation and relevance of the results.

2. Insufficient treatment and interpretation of noble gas, CFC/SF6, and 3H/3He data

The tracer dataset contains internal inconsistencies that remain unexplained:

•Ne concentrations exceed air-saturated water (ASW) values by 121–427%, which is far above typical excess-air ranges expected for such aquifers.

Yes, this is an unusual feature. 7 of 10 He samples were measured in duplicates. Duplicates result in identical data. Because of the large 4He concentrations the amount of gas was split in the measurement system to achieve the proper range for calibration of the signals. This resplitting reduces also the amount of Ne in the device and results in slightly larger errors. While typical error for Ne is about 1% for concentrations near equilibrium, the errors increase to about 1% of value received for the He concentration. I.e., sample "R56 bedrock borehole" with the highest 4He concentration of 1E-1ccSTP/kg created an error for Ne of about 1E-4ccSTP/kg which is exceed by far any plausible concentration. Larger errors for Ne the He concentration beyond these indicate some problems with sampling. The Ne concentrations should not be overrated.

• Some CFC and SF6 values also exceed ASW by large margins.

For the interpretation of CFC and SF6 concentration the amount of excess air is critical. This is usually derived from Ne concentration. Here, this method fails completely and CFC and SF6 data should not be used. Also, SF6 is in general produced in the rock matrix with large rates and should be applied rated here. We have added these notions to the text.

> • Dissolved O2 values reach several hundred percent of ASW (23–401%), including values incompatible with waters sampled at depths up to 160 m.

There were errors in the Table 2. We had mistakenly used the oxygen saturation percentage values for the two "supersaturated" readings of dissolved oxygen. The concentration (mg/l) values are for KUU19 **5.65** and for HÄJY11 **3.14**. The confusion is from the fact that the readings from the YSI meter are displayed both as the concentration and as saturation percentage values for dissolved oxygen and from these two sites, the DO% was marked down to field notes. See also discussion about the high Ne excess.

> • Tritium values are inconsistently reported, and negative 3He* values appear to have been omitted without comment. When applying the unfractionated air (UA) model to the published data, similar 3H/3He results can be reproduced; however, this approach also yields negative 3He* values, which seem to have been excluded from the table without explanation.

Tritiogenic 3He cannot be clearly separated from other 3He sources if 4He concentrations are as large as 5E-4ccSTP/kg. Here data are neglected. Only for samples NOPPA15, KUU19 and HARJA10 3H-3He-age could be derived. We will add a clarifying figure on the relationship between the 3He/4He and the 20Ne/4He ratios in groundwater from the study site in the manuscript.

> • *Such large oversaturations in O2, Ne, CFC, and SF6 could stem from sampling artefacts, air contamination, or analytical issues. If real, they require detailed mechanistic discussion. As presented, the tracer dataset cannot be reliably interpreted.*

See the above discussion. We agree with the reviewer that there were major issues with residence time indicators, and we give explanations to these in the manuscript and advise on careful assessment before using these in the future.

3. Lack of methodological transparency

More detailed information has been added to the Materials and Methods section regarding the field sampling, as well as the calculation of tritiogenic and terrigenic Helium.

4.  Insufficient integration between microbiological and tracer datasets

The redox conditions as well as all samples being hypoxic (there were two incorrect values for DO in Table 2.) at each sampling site are now incorporated into the microbiology discussion, with reflection to the potential cycling of specific elements (C, S, Fe, N). However, as there is no knowledge on the speciation of iron and only total sulphur and SO4 have been measured (sulphite concentrations are lacking), in addition to the limits on making conclusions on microbial metabolic activity from the taxonomy, this somewhat hinders the discussion. Nevertheless, we have now combined the microbial diversity data (Shannon H') with the microbial community structure figures and plotted the relative abundances of known sulphate reducers in samples with the sulphur isotope data to show the relationship between the samples that hosted significant proportions of sulphate-reducers (e.g., *Desulfovibrio*, *Desulfosporosinus*, *Desulfurivibrio*) and the enrichment of the $\delta^{34}S$ values.

The reviewer's notion on the air contamination can be somewhat relevant when considering the DO measurements. These were done in field conditions, pumping the fluids from the aquifer to the surface, and the measurement, although done immediately from the pumped fluids, some oxygen from atmosphere may have dissolved to the fluid. However, as the DO values show hypoxic conditions and most samples are also on the reductive side, we concluded that the groundwater sampled here is most likely oxygen-depleted. Regarding the notion in the reviewer's Figure 3 about the anaerobic fungi, there are several publications that describe diverse fungal communities in hypoxic or anoxic groundwaters in deep terrestrial subsurface as well as subseafloor crust and sediments (Sohlberg et al. 2015, Drake and Ivarsson 2018, Purkamo et al. 2018, Inkinen et al. 2019, Velez et al. 2022). Nevertheless, despite the limited understanding of fungal metabolic capabilities in groundwater systems (Retter et al. 2024), most taxa we identified belong to saprotrophic lineages that are typically adapted to oligotrophic conditions, may be originating from surface as they are distinct from the deeper bedrock groundwater communities such as those described for example in Sohlberg et al. (2015) and Purkamo et al. (2018).

5.  Limited site characterization

Interpreting groundwater flow patterns in this complex, multi-layered aquifer system—partly leaky and partly confined—is challenging outside the recharge zones, which are controlled by bedrock topography. There are two publications describing the conceptual model of the aquifer system referenced (Rashid et al. 2022 and another that has just been published, Åberg et al. 2026 that we can add to the manuscript), and two more on

the pipeline that provide more detailed information on the site. We have now incorporated details on the suspected flow patterns to the Figure 1 and added the references to the papers to the site description part. These have also been incorporated to the discussion in more detail.

6. Figures and tables require substantial revision

We thank the reviewer for providing the guiding figures as we have now a more comprehensive discussion on the end-members based on strontium analyses and the binary mixing trends.

[Figure]

The sample point NOPPA15 is located north of the buried valley site, and the water sample represents a different geochemical environment that just happens to look like a continuation of the blue mixing line. However, it is most unlikely that there is a hydraulic connection between NOPPA15 and the other sampling sites. R56 and HÄJY30 show non-conservative behaviour outside the two mixing lines that suggest three endmembers. The two "outliers" are most likely explained by bedrock groundwater influence in HÄJY30, a groundwater well that has no screen, and R56 being a bedrock groundwater well. This seems to create an additional endmember to the three portrayed in the diagram. In a diagram with the reciprocal of the Sr concentration a binary mixing line draws as a straight line. Here the two dotted lines (red and blue) represent binary mixing trends. The samples near the red dotted mixing line (MIHP15, HÄJY11, MIHP6 and KUU19), represent influence of the Paloluoma buried valley modern/young groundwater. The sample points MIHP15 and MIHP6 are located fairly near the Kyrönjoki valley, but the flow of groundwater in the Paloluoma valley to the north has a volume that overrides the influence from the

Kyrönjoki direction (Fig .1 in the manuscript). The remaining samples on the blue mixing line (LOHI30, HARJA10 and MIETO17) represent the Kyrönjoki valley, south of the buried valley system where there is northbound groundwater flow (figure 1). The mixing line does not clearly represent this south-north direction, due to hydraulic connections west of LOHI30.

As this study was conducted with the purpose of extracting information about the aquifer system prior to extensive groundwater extraction, all the isotope geochemistry results can be considered as end-members.

7. Interpretation remains descriptive

Without a guiding question or conceptual framework, the manuscript does not progress beyond description. A hydrogeological model relating the measured variables to flow, recharge, mixing, or redox evolution would greatly improve interpretive strength. Some figures and questions are added here which might stimulate some interesting scientific discussion.

We thank the reviewer for raising this important point. While building a full numerical or analytical hydrogeological model is beyond the scope of the present study, we agree that the manuscript benefits from a clearer conceptual framework. We have therefore improved the description of aims of the study in the introduction: establishing baseline hydrogeochemical and microbiological properties for future monitoring, determining water origin and potential mixing using isotopes, and assessing groundwater residence time and recharge periods. We also included the need to evaluate the potential of multitracer approach, including the microbial community profiling as one tool to the introduction. We have modified the figure 1 as mentioned above, reorganized and rewrtitten the results and discussion to improve the interpretation and incorporated some additional/improved figures.

Additional suggestion:

We have now had additional help for interpretation and the practices in describing the methods from the expert that made the noble gas analyses in the first place, Jürgen Sültenfuss from University of Bremen. He is listed now as a coauthor.